# Multichannel Classifier for Recognizing Acoustic Impacts Recorded with a phi-OTDR

**DOI:** 10.3390/s23146402

**Published:** 2023-07-14

**Authors:** Ivan Alekseevich Barantsov, Alexey Borisovich Pnev, Kirill Igorevich Koshelev, Egor Olegovich Garin, Nickolai Olegovich Pozhar, Roman Igorevich Khan

**Affiliations:** Photonics and Infra-Red Technology Scientific Education Center, Bauman Moscow State Technical University, 105005 Moscow, Russia; koshelevki@gmail.com (K.I.K.); garinegor@gmail.com (E.O.G.); nickpozhar@gmail.com (N.O.P.); khan.roman.igorevich@yandex.ru (R.I.K.)

**Keywords:** OTDR, classification, acoustics, autoencoder, multichannel signal processing, CNN, signal processing, artificial neural networks, skip connection

## Abstract

The purpose of this work is to increase the security of the perimeter of an area from unauthorized intrusions by creating an improved algorithm for classifying acoustic impacts recorded with a sensor system based on a phase-sensitive optical time reflectometer (phi-OTDR). The algorithm includes machine learning, so a dataset consisting of two classes was assembled. The dataset consists of two classes. The first class is the data of the steps, and the second class is other non-stepping influences (engine noise, a passing car, a passing cyclist, etc.). As an intrusion signal, a human walking signal is analyzed and recorded in frames of 5 s, which passed the threshold condition. Since, in most cases, the intruder moves on foot to overcome the perimeter, the analysis of the acoustic effects generated during the step will increase the efficiency of the perimeter detection tools. When walking quietly, step signals can be quite weak, and background signals can contain high energy and visually resemble the signals you are looking for. Therefore, an algorithm was created that processes space–time diagrams developed in real time, which are grayscale images. At the same time, during the processing of one image, two more images are calculated, which are the result of processing the denoised autoencoder and the created mathematical model of the adaptive correlation. Then, the three obtained images are fed to the input of the created three-channel neural network classifier, which includes convolutional layers for the automatic extraction of spatial features. The probability of correctly detecting steps is 98.3% and that of background actions is 97.93%.

## 1. Introduction

The areas using fiber-optic sensor systems (FOSSs) with distributed sensors are actively expanding. This is due to the creation of new optical devices and the development of electronics and algorithms. Among such algorithms and systems, the methods proposed by A. B. Pnev et al. [1,2,3] have been considered, in which the authors described the propagation of radiation in the inhomogeneous medium of an optical fiber and the possibility of using the created physical model. One priority area is in the protection of a perimeter, wherein a FOSS is used for perimeter security [4,5,6,7,8]. A FOSS based on a phase-sensitive reflectometer was established as an effective automated tool that can help an operator detect intrusions into a protected area. Currently, existing FOSS-based systems contain signal-processing computing modules that enable the use of artificial intelligence algorithms, allowing for a high degree of automation to be achieved.

Convolutional neural networks (CNNs) stand out among the actively used algorithms. For example, in [9] by Yi Shi et al., the resulting space–time diagram matrices are filtered by three different bandpass filters to extract event features and form RGB images, where they are then fed into the Inception_V3 network. Five types of events are classified. The accuracy is 86%. In [10], Hongqiao Wen et al. use a CNN to perform a spectrogram analysis. The problem to be solved is the classification of five types of events. The recognition accuracy is 80%. In [11], Suzhen Li et al. use a CNN to identify third-party threats to buried municipal pipelines. The paper [12,13] considers machine learning methods for identifying and classifying various events. In [14], H. Salem et al. studied the optimal use of CNN architecture for the image classification of various ocular diseases. The use of recurrent neural networks (RNN) makes it possible to reveal hidden patterns in the signals recorded by a reflectometer since they depend on time. Zhandong Wang et al. [15] use an architecture similar to ours in that it uses several channels for data processing, but instead of CNN, LSTM is used, and the input data is presented as a one-dimensional vector. The works [16,17] also analyze time dependencies using recurrent networks.

Considering the human factor, an operator may gain confidence in the predictions of security system algorithms. Since classification algorithms may not provide accurate predictions, intrusions may be missed. Therefore, there is a need to refine and create new algorithms for recognizing sources of acoustic influences to reduce the likelihood of false alarms and the omission of signals related to penetration actions. One such algorithm is described in the work by I. A. Barantsov et al. [18], in which the researchers created a neural network classifier architecture using a CNN to automatically extract features from segments of a space–time diagram, registering acoustic fields. The desired signals were images of acoustic influences generated during a human step. The disadvantages of this algorithm included the proposed architecture of the neural network not being able to completely select all the necessary spatial features since noise and artifacts that were not related to the signal could interfere with its recognition. Increasing the depth of the neural network would not increase the recognition efficiency, given the low spatial hierarchy and low requirements for the invariance of the signal images. Moreover, a desirable feature of a perimeter detection system is an increased sensitivity in detecting and classifying weak impacts (e.g., quiet footsteps). With a quiet step, the recorded signals have low energy, which worsens their recognition. Thus, this work is aimed at creating an algorithm that allows for the classification of acoustic effects with different energies generated during human steps with high accuracy. Various feature extraction algorithms have a number of advantages and disadvantages. Therefore, multichannel image processing [19,20], which takes one image as input, processes it with several different feature extraction algorithms, and returns several processed images, is an effective solution for feature extraction. The creation of neural network classifier architecture that has several inputs for processed images in order to extract spatial features from each image independently of others is also an important point for achieving high-accuracy predictions.

## 2. Materials and Methods

### 2.1. Input Data

The main optical module that allows for the registration of acoustic waves is phi-OTDR, the purpose of which is to register backscattered light. In this work, scattered Rayleigh light from scanning pulses was recorded, which had a wavelength that was equivalent to incident light (Figure 1).

The detected radiation was the sum of the amplitudes, taking into account the phase of the backscattered waves; therefore, we investigated their interference [21,22]. The phase of the reflecting centers changed when exposed to a certain section of the fiber, which affected the interference of the scattered waves and led to changes in the power received at the receiver. Systems of this type are sensitive to environmental vibrations; for example, the moment of each step of a human walking is a source of acoustic influences. The generated acoustic waves exert pressure on the fiber-optic cable; depending on the parameters of the acoustic impact, the refractive index is modulated. That is, the position of particles in the crystal lattice along the axis of action relative to each other will change with the same frequency as the incoming wave. In the case of a sinusoidal acoustic effect, the change in the grating would look like the following figure (Figure 2).

Thus, the received signal traces contain information about the source of the acoustic influences. If the signal traces are recorded at a frequency of 1000 Hz and concatenated according to the recording time, then a space–time diagram is obtained (Figure 3), where the *x*-axis is the axis of the space, the maximum value of which is equal to the length of the sensor cable; the *y*-axis is the time axis; and the color (pixel brightness) indicates the intensity of the acoustic impact.

The horizontal axis represents the spatial length and can be used to determine at what point in the fiber the acoustic impact was recorded. The maximum value of the unscaled *x*-axis, taking into account the spatial resolution, represents the length of the fiber.

Here, the signal traces accumulated for 5 s, after which, the segments measuring 5000 × 150, selected on the basis of the threshold condition, were allocated. The selected segments were thinned along the 1/100 time axis, so the size of the input data was 50 × 150 (Figure 4).

### 2.2. Feature Extraction Algorithms

#### 2.2.1. Denoising and Artifact Removal

The resulting diagrams contained information about the spatial distributions of the recorded acoustic fields. Since the propagation medium was non-uniform, and the optical fiber had manufacturing defects, there was a DC component at different points in space. Moreover, the value of the constant component was influenced by the low-frequency background, which had a heterogeneous nature of occurrence. The pressure of the searched acoustic impact exerted on the sensor cable was not a constant value and, therefore, not a stationary process in the selected segments. Thus, subtracting the stationary component at each point in space was an effective solution for removing information that was not related to the desired signals. A linear trend was calculated along the time axis for each spatial sample (1):(1)$$y_{t}=a_{0}+a_{1}t$$

The above equation represents a straight line, in which the trend parameters were calculated using the least squares method. The values $a_{0}$ and $a_{1}$ were determined from the minimum condition function (2)
(2)$$f\left(a_{0},\;a_{1}\right)=\sum_{t=1}^{n}{\left(y_{t}-a_{1}t-a_{0}\right)}^{2}$$

After this, a mask of the stationary component was created (Figure 5b), which was then subtracted from the original image. Furthermore, to reduce the data dimension, elements whose values were less than zero were set to zero (Figure 5c). This operation did not entail the loss of important information.

The results of this operation contained information of interest, but there were various noises and artifacts in the image (3), such that
(3)$$I_{r}\left(m,n\right)=I_{s}\left(m,\;n\right)+I_{µ}\left(m,\;n\right)$$
where $I_{s}\left(m,\;n\right)$ represents the values of the desired signals; $\;I_{µ}\left(m,\;n\right)$ represents the values of the noise components and artifacts; and $\;I_{r}\left(m,n\right)$ represents the value of the registered signal, which negatively affects the recognition of the desired signals. Given that component $I_{µ}\left(m,\;n\right)$ has a complex nature of occurrence, the spectral density of which can be greater than the spectral density of the desired signals, using a linear filter will not produce an effective result.

An autoencoder is a type of neural network whose purpose is to copy the features of the input to the output. The autoencoder first encodes the input image into a smaller hidden representation and then decodes it back into an image. The work by Francois Chollet et al. [23] described the operation of autoencoders for image processing. This consisted of two parts: the encoder, h = f(x), and the decoder, r = g(h). If the autoencoder is trained to set g(f(x)) = x everywhere, this will not be useful for solving the problem at hand. Therefore, autoencoders are designed in such a way that they cannot learn to copy image data perfectly. Since autoencoder models are forced to prioritize which input data elements to copy, they often extract useful input data. An autoencoder learns to compress the data, minimizing reconstruction errors. Similar studies were carried out by Abdelli K. et al. [24,25]. In their work, an autoencoder with the most suitable architecture (Figure 6) was trained to solve the problem of noise reduction and artifact removal.

The optimization criterion was the cross-entropy (4):(4)$$H\left(P,\;Q\right)=-\sum_{x}P\left(x\right)logQ\left(x\right)$$
where *P* represents the distribution of the labels, and *Q* represents the probability distribution of the model predictions.

The difference between two probability distributions is measured by their cross-entropy. If the cross-entropy value is small, the distributions are similar.

The quality of the autoencoder was evaluated visually. The results of the created model are shown in Figure 7.

#### 2.2.2. Highlighting Periodic Signal Structures

One feature of a desired signal is its periodic structure. Background components may be similar to signal segments, but they may not be periodic.

The most-used operation to extract periodicity is correlation; however, it has significant disadvantages. The signal energy decreases depending on the distance of the coordinates from the central one, thus causing the signals at the edges to be poorly distinguished. In this paper, a different approach to calculating correlation is proposed. Consider the formula for calculating Pearson correlation coefficients (5):(5)$$r_{ij}=\frac{\sum_{m}\left[I_{1}\left(i+m\right)-\bar{I_{1}}\right]\left[I_{2}\left(i+m\right)-\bar{I_{2}}\right]}{\sqrt{\sum_{m}{\left[I_{1}\left(i+m\right)-\bar{I_{1}}\right]}^{2}\cdot \sum_{m}{\left[I_{2}\left(i+m\right)-\bar{I_{2}}\right]}^{2}}}$$
where $I_{1}$ denotes the first input image; $I_{2}$ denotes the second input image; $r_{ij}$ denotes the Pearson correlation coefficient; and $\bar{I}$ denotes the mean of the image.

Coefficients are calculated along the time axis, so the formula has one variable. The parameters of this formula are the mean value and standard deviation, which are constant values, but they can be used adaptively. The adaptability lies in the fact that the mean value and standard deviation are calculated at each iteration. Thus, we get the adaptive correlation formula (6):(6)$$r_{ij}=\frac{\sum_{m}\left[I_{1}\left(i+m\right)-A\right]\left[I_{2}\left(i+m\right)-B\right]}{\sqrt{\sum_{m}{\left[I_{1}\left(i+m\right)-A\right]}^{2}\cdot \sum_{m}{\left[I_{2}\left(i+m\right)-B\right]}^{2}}}$$
where $A=\frac{\sum_{m}I_{1}\left(i+m\right)}{m}\;and\;B=\frac{\sum_{m}I_{2}\left(i+m\right)}{m}$ are the adaptive correlation parameters.

The results of the adaptive correlation are shown in Figure 8c.

Despite the noise component, the results of the adaptive correlation depicted restored segments related to the periodic structure of the signal.

### 2.3. Multichannel Neural Network Classifier

To find a larger number and better quality of features, architecture was created that allowed for the extraction of the most suitable features from each channel of the input tensor (Figure 9). The first channel contained the selected raw segment data; the second contained the segment processed by the autoencoder; and the third contained the results of the adaptive correlation.

Since the designed algorithm works with spatial features, convolutional layers were actively used in the architecture [26,27]. The convolution layer contained a certain number of kernels, the values of which were weight coefficients. These are called convolutional layers because the basic operation is a discrete convolution of two signals. Convolution should be understood as a mathematical method of combining two signals to form a third signal, as described by Formula (7):(7)$$y\left(k,\;l\right)=f\left(k,\;l\right)*g\left(k,\;l\right)=\sum_{m=-\infty }^{\infty }\sum_{n=-\infty }^{\infty }f\left(m,\;n\right)g(k-m,\;l-n)$$
where $f\left(m,n\right)$ indicates the first input image, and $g\left(m,n\right)$ indicates the second input image.

The rectified linear unit (ReLU) function was used as an activation function because of its simplicity, eliminating the need to use more difficult activation functions which could reduce the accuracy of the classifier predictions. The input tensor was divided into 3 channels, in which the unique feature characteristics of each image were extracted. Then, the channels were combined into one tensor, building the convolutional layers. In [16], it was mentioned that the desired signals in the images of a space–time diagram have low requirements for invariance and a fairly low spatial hierarchy; therefore, 3 convolutional layers were enough to extract features from them. The use of skip connection technology made it possible to add 2 more layers. This allowed for the extraction of deeper spatial features without losing the accuracy of the classifier. Pooling layers were used in combination with convolution layers (Bieder, F. et al. [28]) to reduce the size of a tensor. In this paper, we propose the use of max pooling, which samples from 4 neighboring elements since the signal image consists of light stripes on a dark background, making them more perceptible. After passing through the max-pooling layer, the number of data is decreased by a factor of four, considerably reducing the computational cost of training without sacrificing crucial information.

Then, the data were flattened into a vector, and the dropout operation [29] was performed, the purpose of which was to reduce overfitting and reduce the probability of falling into a local minimum during training. After that, the vector entered the fully connected layers (FC), the purpose of which was to find hidden features. The two outputs of the last fully connected layer provided data to the softmax activation function, which then returned probabilistic class membership values in the range [0, 1].

The criterion for optimizing a neural network was to minimize the value of the categorical cross-entropy.

Adam (adaptive moment estimation) was used as an optimizer since it can cope best with local minima and overcome the complex landscape of the objective function to solve the optimization problem.

## 3. Experimental Study and Discussion

φ-OTDR is a device capable of detecting acoustic effects on a cable on the basis of Rayleigh backscatter analysis (Stepanov, K.V. et al. [30]). The scheme of the device is shown in Figure 10. The light source used here was a frequency-stabilized laser with a narrow spectral linewidth, the coherence length of which was much greater than the pulse half-width (τ_pulse ~10–500 ns). This caused interference in the backscattered radiation at each pulse position. The continuous laser beam was amplified by an erbium-doped fiber amplifier (EDFA). After the acousto-optic modulator (AOM) modulated the radiation into probing pulses, it entered the sensor fiber through a circulator in the forward path. The backscattered radiation traveled in the circulator in the opposite direction. It was then amplified by the EDFA. Enhanced spontaneous emission (ASE) was eliminated after passing through a narrowband filter. Next, the radiation entered a photodiode (PD) and was digitized with an analog-to-digital converter (ADC) before being processed on a personal computer (PC).

The experiments on the registration of acoustic signals along a distributed sensor with a length of 40 km are described in this paper, and the data of the step signals and non-step-related signals are obtained. Figure 11 shows a block diagram of the designed classification algorithm.

Before the classifier model was trained, a neural network denoising autoencoder was trained. Noisy and non-noisy data were used in the autoencoder training. It was impossible to obtain experimentally noise-free data because the radiation receiver registered a mixture of signal, noise, and artifacts. Therefore, because our dataset consisted of a set of noisy images, we manually removed unnecessary artifacts and noises using image editors (Figure 12). The operator had extensive experience in conducting experiments and was able to recognize the signal with high accuracy. Thus, accidentally erasing signal elements would not affect the operation of the encoder since such actions were not systematic.

The dataset comprised 1500 real and 1500 processed images. The results of the trained autoencoder can be seen in Figure 6. When training the autoencoder, loss curves were obtained (Figure 13).

On the basis of a priori information about the events, the data for training the classifier were extracted to create a dataset and saved. Then, augmentation was carried out on the existing dataset. For the experiment, 5000 false and 5000 step signal data were obtained. The sets of training and validation data were divided in a ratio of 3:1 (Vrigazova, B. et al. [31]), respectively. The number of test data is 2800 (1400 false and 1400 steps signal data). The optimal number of epochs was 120. When training the multichannel classifier, loss curves were obtained (Figure 14).

Accuracy curves were also obtained (Figure 15).

The classification accuracy on the test set was greater than 98.1%.

Models based on popular architectures such as DenseNet169 (Figure 16a), EfficientNetB7 (Figure 16b), InceptionV3 (Figure 16c), Xception (Figure 16d), ResNet50 (Figure 16e), and a previously developed architecture (Figure 17) were trained on the original dataset. This makes it possible to compare the obtained results of the algorithm proposed in the manuscript and popular architectures. Hyperparameters and additional layers to the basic architectures were selected experimentally in such a way as to achieve the highest prediction accuracy.

Figure 18 shows learning curves for each compared model.

The error matrix was also calculated (Table 1).

Based on the results obtained, we can conclude that the accuracy of the created algorithm is superior to the others. Analyzing the results of confusion matrices, one can see that the accuracy of predictions of some models for one of the classes exceeds the accuracy of the created algorithm; however, the accuracy of predictions for another class is unacceptably lower, which makes it impossible to use them in perimeter detection systems.

The compared models based on different architectures were trained and tested on the same training, validation, and test datasets, which excludes the influence of the experimental setup on the discrepancy between the results. The developed algorithm differs in that three images are received at the input of its channels, which are processed using an autoencoder and adaptive correlation. In the channels, the unique features of each image are extracted, regardless of others, which is not implemented in popular architectures. It is also worth noting that most models have a large number of layers to extract features in images that have a large depth of spatial hierarchy. The analyzed signal images have a simple texture.

Because neural networks are especially difficult to interpret, one of the important and interesting points is to create an explanation of the basis on which the trained model makes a particular decision. Since the designed architecture contains convolutional layers, the analysis of weight coefficients is one of the effective ways to interpret the results. We are able to visualize the patterns on which the filters in the CNN provide the greatest response. To obtain patterns, it is necessary to solve a similar problem as in training—fit the values of the image so that it provides the greatest possible response when convolved with a kernel from the convolutional layer. Thus, each core will have its own pattern (Figure 19).

It can be seen that some filters in this layer of the neural network, which detect features in images with human step signals, are looking for a periodic structure, but since the signals are invariant to the coordinates in the image, the structure of the pattern may look like a noisy one.

The created algorithm works with the spatial features of signal images, but it does not take into account the analysis of series along the time axis in order to identify temporal dependencies. One of the priority areas for improving this algorithm is the addition of time series processing.

## 4. Conclusions

A threshold condition was used to extract useful information from the space–time diagrams of acoustic fields recorded with a FOSS based on a φ-OTDR and select two-dimensional 50 × 150 segments. A denoising autoencoder model was created alongside a training dataset that included segments of recorded human steps and segments that were processed using an image editor. A mathematical apparatus was created to perform adaptive correlation without a loss in signal energy, depending on the distances of the coordinates from the central one; this made it possible to effectively isolate the periodic components of the signal. The desired images of acoustic signals (human steps) and the desired images of various acoustic backgrounds were combined to create a dataset with two classes. The dataset was then expanded using augmentation. A model of a multichannel classifier was created using CNN layers, which allowed for the selection of optimal features from each channel. The optimal number of training epochs and the learning rate were chosen on the basis of the trained neural network’s learning curves. As a result, a classifier with sufficiently high prediction accuracy was obtained. Comparing the forecast accuracy estimates and the confusion matrix obtained during experiments with neural network models based on the created algorithm, popular architectures, and previous neural network architecture, we can conclude that a sufficiently high forecast accuracy was achieved, which made it possible to increase the accuracy, including during operation with weak impact.

## Figures and Tables

**Figure 1 sensors-23-06402-f001:**
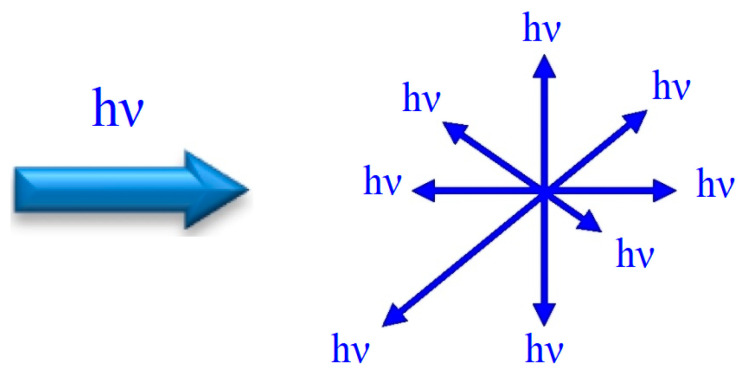
Diagram of the Rayleigh scattering process.

**Figure 2 sensors-23-06402-f002:**
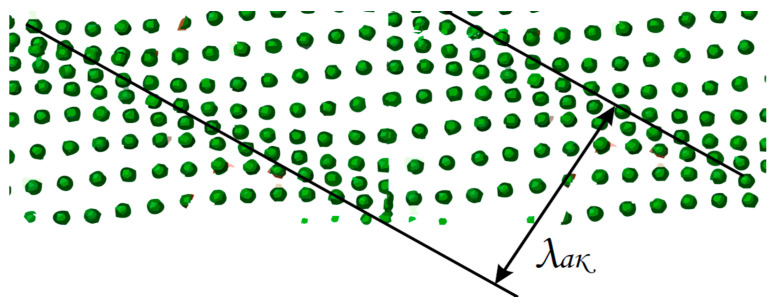
Dependence of the position of particles relative to each other on the frequency of an acoustic impact ($\lambda_{ak}$ is the wavelength of the acoustic impact).

**Figure 3 sensors-23-06402-f003:**
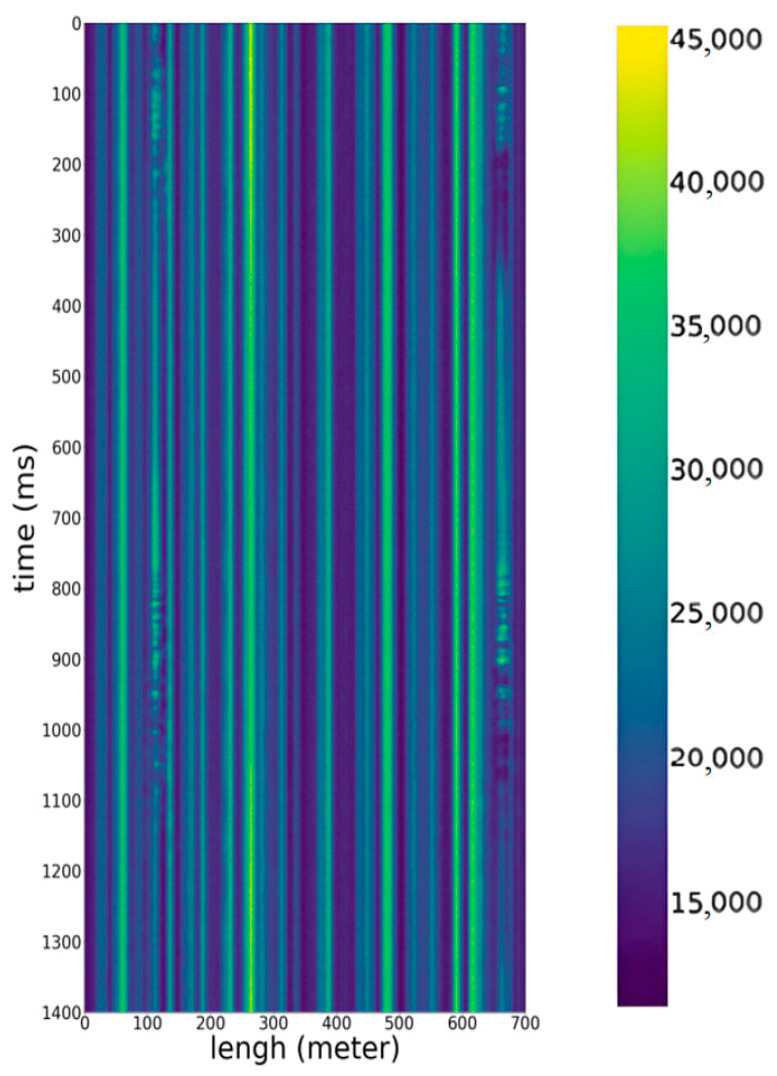
Visualization of the space–time diagram image.

**Figure 4 sensors-23-06402-f004:**
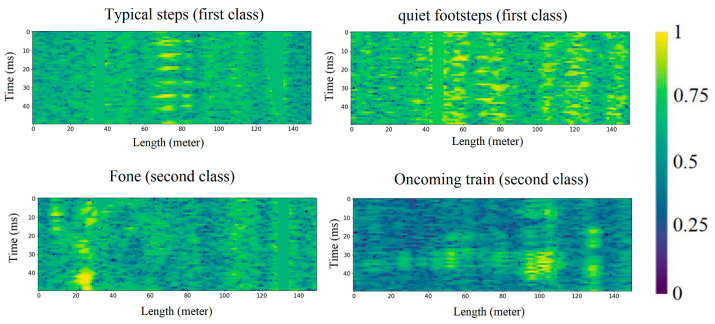
Dataset examples.

**Figure 5 sensors-23-06402-f005:**
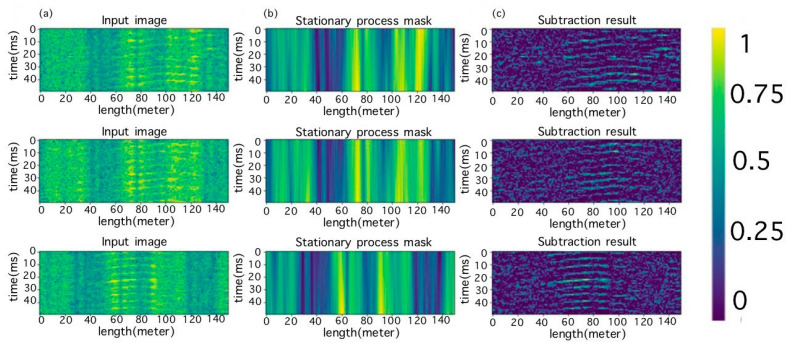
(**a**) The selected segment, (**b**) the resulting masks of stationary components, and (**c**) the results of subtraction.

**Figure 6 sensors-23-06402-f006:**
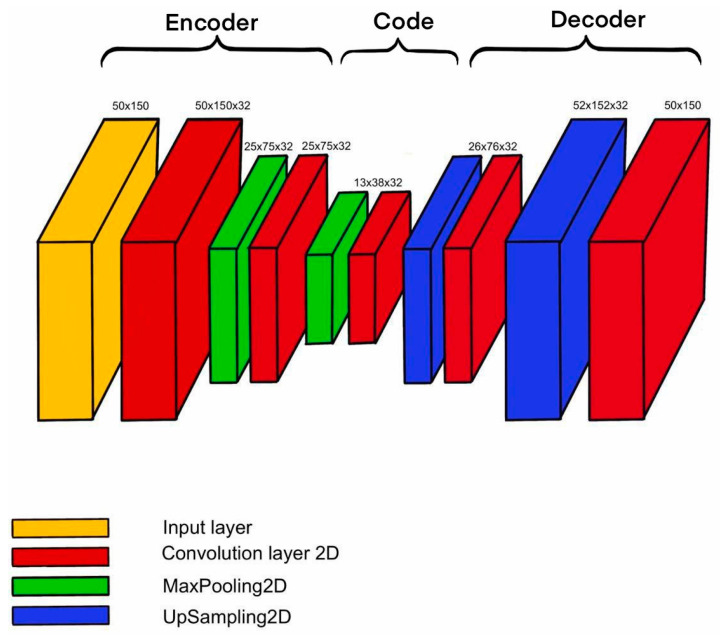
The architecture of the denoising autoencoder.

**Figure 7 sensors-23-06402-f007:**
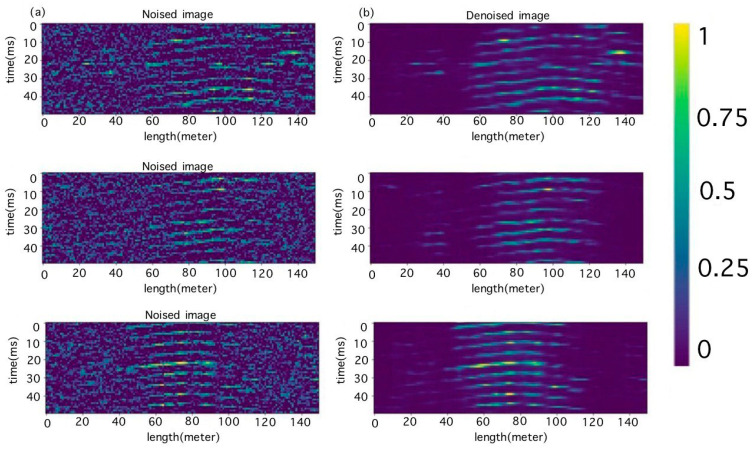
(**a**) Noisy test images and (**b**) results of autoencoder processing.

**Figure 8 sensors-23-06402-f008:**
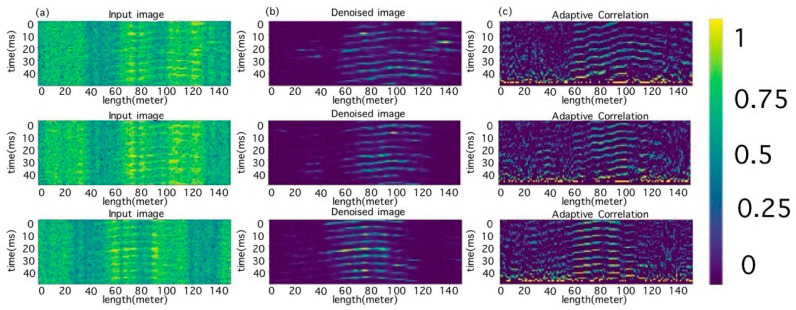
(**a**) Original image; (**b**) the results of autoencoder processing; and (**c**) the results of adaptive correlation.

**Figure 9 sensors-23-06402-f009:**
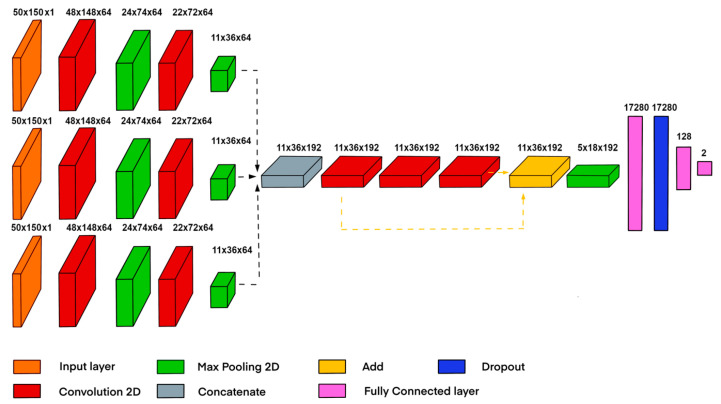
Architecture diagram of the multichannel classifier.

**Figure 10 sensors-23-06402-f010:**
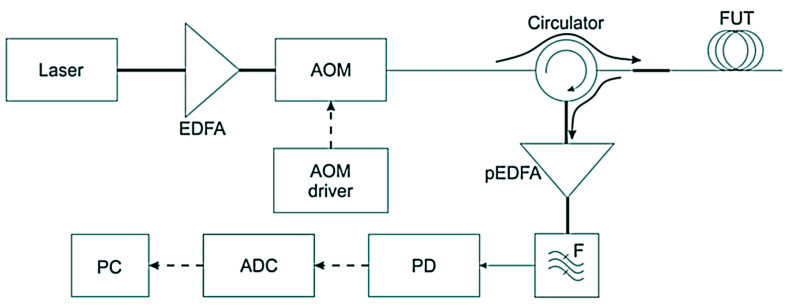
Experimental setup of the system based on φ-OTDR.

**Figure 11 sensors-23-06402-f011:**
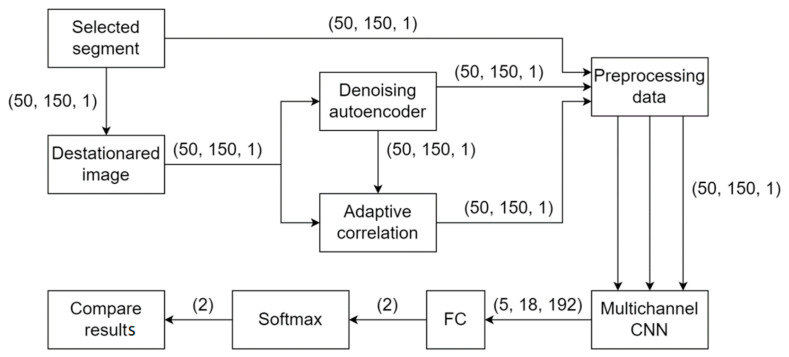
General block diagram of the developed algorithm.

**Figure 12 sensors-23-06402-f012:**
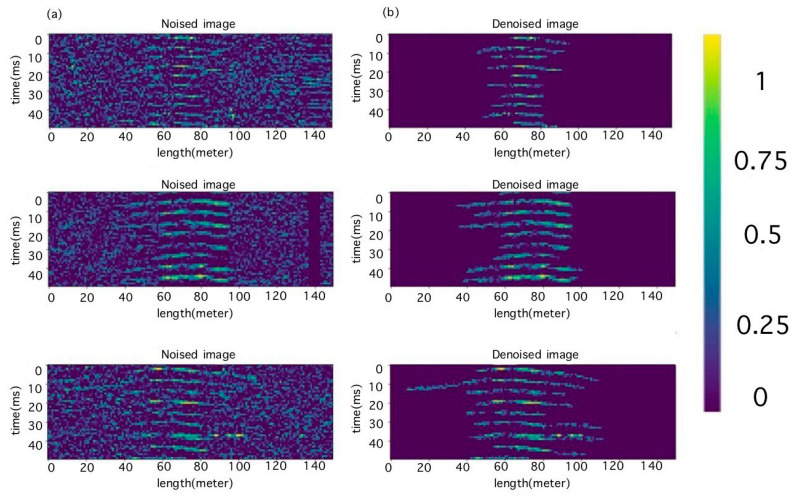
Examples of (**a**) noisy and (**b**) processed images from the dataset.

**Figure 13 sensors-23-06402-f013:**
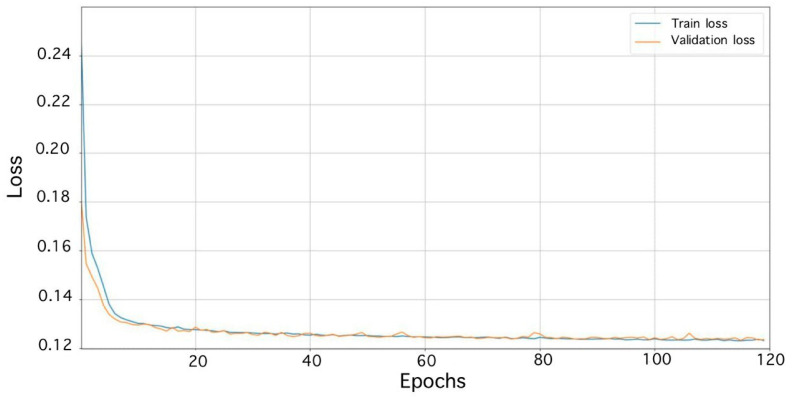
Learning loss curves of the denoising autoencoder.

**Figure 14 sensors-23-06402-f014:**
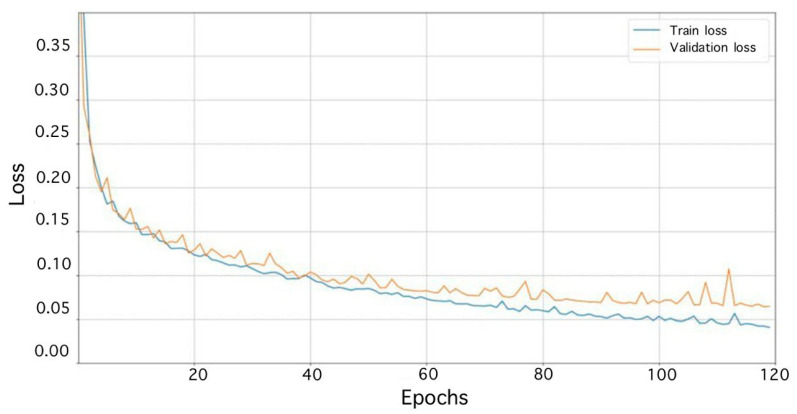
Learning loss curves of the multichannel classifier.

**Figure 15 sensors-23-06402-f015:**
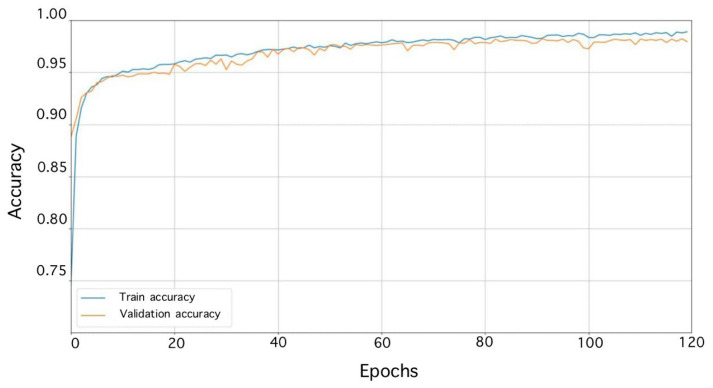
Learning accuracy curves of the multichannel classifier.

**Figure 16 sensors-23-06402-f016:**
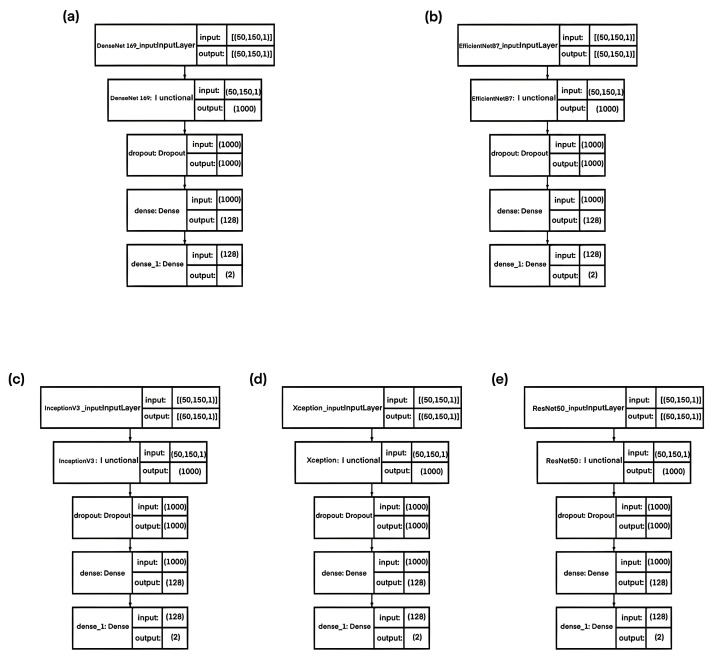
Architectures of neural network classifiers for comparison: (**a**) based on DenseNet169; (**b**) based on EfficientNetB7; (**c**) based on InceptionV3; (**d**) based on Xception; (**e**) based on ResNet50.

**Figure 17 sensors-23-06402-f017:**
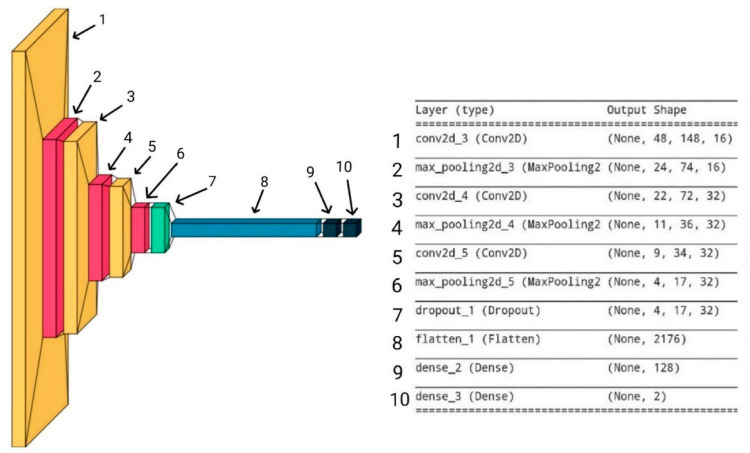
Neural network architecture (described in [18]).

**Figure 18 sensors-23-06402-f018:**
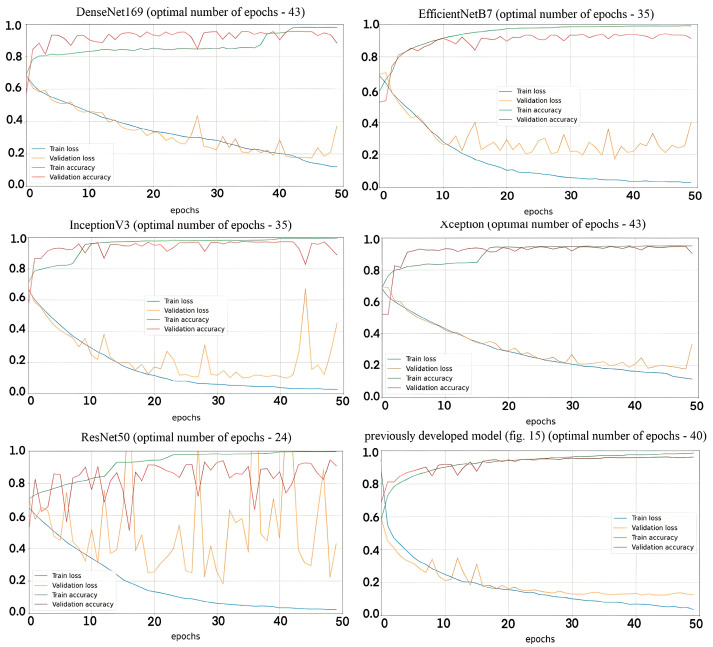
Graphs with learning curves of compared models.

**Figure 19 sensors-23-06402-f019:**
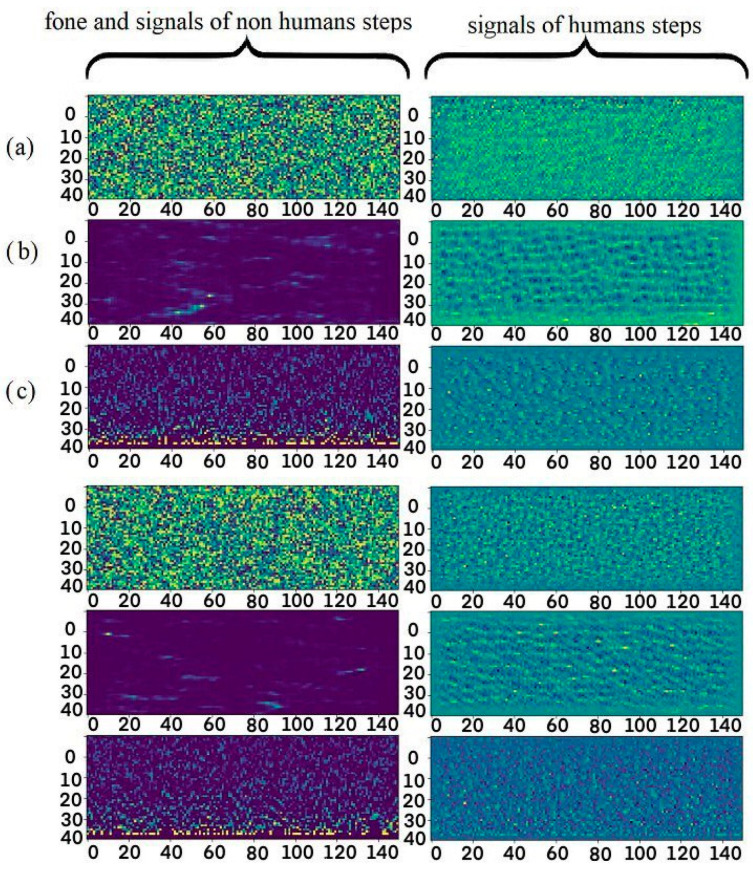
Visualization of patterns on which some filters in the last layer provide the maximum response (**a**) for the input of the first channel, (**b**) for the input of the second channel, and (**c**) for the input of the third channel.

**Table 1 sensors-23-06402-t001:** Results of evaluating the quality of predictions using various metrics.

Classificator	Accuracy	Confusion Matrix		
			Actual Positive	Actual Negative
Created algorithm (Figure 10)	98.1%	Predicted positive	98.3%	1.7%
		Predicted negative	2.07%	97.93%
DenseNet169 (Figure 16a)	90.17%	Predicted positive	99.04%	0.96%
		Predicted negative	18.7%	81.3%
EfficientNetB7 (Figure 16b)	91.3%	Predicted positive	98.53%	1.47%
		Predicted negative	12.58%	87.42%
InceptionV3 (Figure 16c)	96.7%	Predicted positive	95.12%	4.88%
		Predicted negative	1.71%	98.29%
Xception (Figure 16d)	93.17%	Predicted positive	98.84%	1.12%
		Predicted negative	12.5%	87.5%
ResNet50 (Figure 16e)	83.96%	Predicted positive	71.7%	28.3%
		Predicted negative	3.79%	96.21%
Previously developed architecture (Figure 17)	96.07%	Predicted positive	96.44%	3.56%
		Predicted negative	4.3%	95.7%

## Data Availability

Not applicable.

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
