# Peer review of "Multichannel Classifier for Recognizing Acoustic Impacts Recorded with a phi-OTDR"

_sensors, 2023, doi:10.3390/s23146402_

Round 1
Reviewer 1 Report
The authors claimed that they have proposed Multichannel Classifier for Recognizing Acoustic Impacts Recorded with a phi-OTDR. This is an interesting work, but before publication authors need to clarify some points given below:
1. First of all, the authors are suggested to improve the descriptions of the abstract and highlight the impact of this work.
2. What does λak represent in Figure 2 ?
3. What are the units of the horizontal and vertical coordinates in Fig.12-Fig.14? and What does the horizontal and vertical coordinates represent?
4. Does the horizontal axis in Fig. 3 represent the fiber length or spatial length? Please provide a detailed explanation.
5. Please add a comparison of the research results with other studies to illustrate the advantages of this study.
Moderate editing of English language required
Reviewer 2 Report
Dear Authors,
I have reviewed the manuscript “Multichannel Classifier for Recognizing Acoustic Impacts Recorded with a phi-OTDR” Manuscript ID: sensors-2438367 that has been submitted for publication in the: Sensors (ISSN 1424-8220) and I have identified a series of aspects that in my opinion must be addressed in order to bring a benefit to the manuscript.
The article under review will be improved if the authors address the following aspects in the text of the manuscript:
1. The abstract does not provide sufficient background information about the problem being addressed or the significance of solving it. It would be helpful to include details such as why distinguishing between human footsteps and other sources of acoustic waves is important and what potential applications or implications this algorithm may have.
2. The abstract states the prediction accuracy of test data but does not elaborate on the evaluation process or metrics used. Providing more information about the dataset, experimental setup, and additional performance measures (e.g., precision, recall, F1 score) would enhance the credibility of the results.
3. The section “related work” is needed and make a table with the latest research that has been done.
4. The introduction’s “last section” does not explicitly mention the specific contributions of the study, which is one of the weak aspects identified. And the definition of sections in the rest of the paper.
5. “Figure 5. The architecture of the denoising autoencoder”, this figure needed to be enhanced and given more details.
6. It is preferable to use more than one dataset, at least 3 of the datasets, in order to ensure the results of the proposed framework used.
7. Machine learning algorithms, especially complex ones, can often lack interpretability. Understanding the underlying factors or features driving the predictions is important in a clinical setting for physicians to make informed decisions. If the developed model lacks interpretability, it may limit its practical utility and acceptance in the medical community.
8. In addition to comparing the results with previous research, it is important to include baseline models or methods for comparison. These baselines should be representative of the state-of-the-art or commonly used approaches in the field. Comparing against baselines helps establish the effectiveness and superiority (if applicable) of the proposed method.
9. There are discrepancies in the results between the paper and previous research, it is essential to discuss and analyze the potential reasons behind these differences. Factors such as variations in the experimental setup, feature selection, data preprocessing, or model architectures could contribute to disparities. Addressing these differences helps in understanding the unique aspects and limitations of the proposed approach.
10. The references need to be updated for the years 2022 and 2023, as this field has been recently raised.
DOI: 10.1109/NRSC49500.2020.9235095
Recognition of Ocular Disease Based Optimized VGG-Net Models
11. It would be beneficial to include a statement about potential future directions or improvements for the proposed algorithm. This could help in identifying the limitations and opportunities for further research and development.
Reviewer 3 Report
This manuscript reports multichannel classifier for acoustic impact with a phi-otdr. I think that the manuscript can be accpeted after the following issues are solved.
1, As the prediction accuracy is 98.1%, what is the conventional accuracy for this accoustic signal?
2, What motivation for using the current method in Introduction? Any reason? The author can cite more papers about method to support the idea in introcution section, such as Spectrum analysis enabled periodic feature reconstruction based automatic defect detection system for electroluminescence images of Photovoltaic modules. Micromachines. 2022, 13, 332.
3, How compared the current method to other exsiting method? The results look not that good compared the other exsting method. What is the noverlty of this one?
4, What is the reason to choose multichannel classifier?
5, Please identify the unit of y axle in all figures in this manuscript
It is fine to understand
Reviewer 4 Report
The article is original and of interest, but has design flaws. For the captions in Fig. 7 and 15 used a small font size, they are hard to read. The title of article 19 in the reference is presented in Russian. The authors should increase the font size for the captions in Figures 7 and 15. The title of the article 19 should be translated into English.
Round 2
Reviewer 2 Report
Accept in present form
Reviewer 3 Report
The authors did solve all my issues. I recomment to accept in the current form.
It is fine for readers to read.